# Significance of Lipid Fatty Acid Composition for Resistance to Winter Conditions in *Asplenium scolopendrium*

**DOI:** 10.3390/biology11040507

**Published:** 2022-03-25

**Authors:** Alexander Voronkov, Tatiana Ivanova

**Affiliations:** K. A. Timiryazev Institute of Plant Physiology, Russian Academy of Sciences, IPP RAS, 35 Botanicheskaya St., 127276 Moscow, Russia; itv_2006@mail.ru

**Keywords:** *Asplenium scolopendrium*, medium-chain fatty acids, arachidonic acid, eicosapentaenoic acid, cold stress

## Abstract

**Simple Summary:**

Plants growing at temperate and polar latitudes are exposed to cold stress. With climate change, different durations of low temperatures and sometimes frost are being increasingly observed in regions at low latitudes. This can cause especially great harm to agricultural crops, the mortality of which can pose a serious challenge to the food supply for the global population. One of the factors affecting plant resistance to low and negative temperatures is specific changes in the fatty acid (FA) composition of lipids. It should be noted that most of the crops studied in this regard are angiosperms. It is known that the FA composition of angiosperms has undergone significant evolutionary changes compared to that of nonflowering vascular plants. Studying the FA composition of various taxonomic groups can shed light on and reveal new mechanisms of plant resistance. Therefore, in this paper, we focused on the rare evergreen fern *Asplenium scolopendrium*, whose fronds can tolerate freezing. A number of specific features of its FA composition were discovered, which, in combination with other resistance mechanisms, determine its ability to grow in temperate climate zones and safely undergo wintering.

**Abstract:**

Ferns are one of the oldest land plants. Among them, there are species that, during the course of evolution, have adapted to living in temperate climates and under winter conditions. *Asplenium scolopendrium* is one such species whose fronds are able to tolerate low subzero temperatures in winter. It is known that the resistance of ferns to freezing is associated with their prevention of desiccation via unique properties of the xylem and effective photoprotective mechanisms. In this work, the composition of *A. scolopendrium* lipid fatty acids (FAs) at different times of the year was studied by gas–liquid chromatography with mass spectrometry to determine their role in the resistance of this species to low temperatures. During the growing season, the polyunsaturated FA content increased significantly. This led to increases in the unsaturation and double-bond indices by winter. In addition, after emergence from snow, medium-chain FAs were found in the fronds. Thus, it can be speculated that the FA composition plays an important role in the adaptation of *A. scolopendrium* to growing conditions and preparation for successful wintering.

## 1. Introduction

Ferns were one of the first vascular plants that, because of the development of the xylem, successfully settled on Earth’s surface as early as the Palaeozoic era [1]. For many millions of years, they were the dominant plant form on land [2]. However, later, in the Mesozoic, along with a gradual cooling of the climate, gymno- and angiosperms gradually replaced ferns as the most abundant plants [3,4,5]. However, ferns were able to successfully coexist with angiosperms and continued to evolve [6]. They occupy the undergrowth [7], and many ferns have become epiphytes [8]. Thus, ferns now occupy ecological niches with conditions different from those during the period of their dominance. They had to adapt to a lack of moisture and low light levels. In addition, some fern species have successfully adapted to temperate conditions [9], and some species are even truly arctic, subantarctic, and alpine [10,11].

Among temperate ferns, some species retain their fronds in winter [12,13]. It is known that such ferns are able to tolerate negative temperatures, with some species tolerating temperatures as low as −20 °C [14]. Thus, these ferns tolerate repeated freeze-dry cycles [15,16,17,18]. There is a theory that some species of ferns, in particular species of *Asplenium*, may cope with winter stress via poikilohydric behavior [19]. Plant cells with desiccation tolerance (DT) have anatomical and biochemical mechanisms that facilitate the dehydration process, including metabolic arrest, with successful subsequent restoration of their activity during rehydration [20]. The main features of vegetative DT are photoprotection against redox processes due to low-molecular-weight antioxidants and the ability to preserve the integrity of cells (for example, cell walls and membranes) [21,22]. Vegetative tissue DT is very rare in angiosperms and is also rare but more common (approximately 10 times) among ferns [23,24]. At the same time, it has long been known that for species with DT, resistance to freezing increases significantly under arid conditions [25].

The recent work of Fernández-Marín et al. [26] was focused on the ability of fern fronds to withstand low temperatures and freezing. The authors showed that adaptation for freezing tolerance is associated with DT through complementary xylem properties (which may prevent risk of irreversible cavitation) and effective photoprotection mechanisms. However, the lipid fatty acid (FA) composition of the membrane was not considered by the authors as an important parameter for plant resistance and adaptation.

FAs are considered one of the general plant defense systems against various biotic and abiotic stresses [27]. The protective function is performed mainly by unsaturated FAs, which, in plant cells, act as components and modulators of cellular membrane glycerolipids, stocks of extracellular barrier constituents, precursors of various bioactive molecules, and regulators of stress signaling. Several recent reviews [28,29,30,31] have been devoted to all of these aspects, but in our work, we focused on the role of FAs in plant cold stress.

It is known that the lipid FA composition in plant membranes plays an important role in adaptation to low and subzero temperatures [32,33]. Previous studies reported that the accumulation of unsaturated FAs ensures the safety of winter wheat seedlings in winter [34,35]. The accumulation of unsaturated FAs in cell membranes, especially that of polyunsaturated FAs, promotes plant resistance to damage by low temperatures [36,37]. The synthesis of unsaturated FAs in plant cells occurs with the participation of desaturases [38]. All genes encoding FA desaturases (FADs) exhibit cold-induced and heat-repressed expression, although with distinct regulatory time courses, which indicates the potential roles of FADs in temperature stress resistance [39]. However, the FA composition of the resistance of ferns to winter conditions is practically unstudied. Dehydration leads to damage to cell membranes [40,41,42], and FAs contribute to better adaptation of cells to dehydration due to their role as components of membrane lipids [43]. Based on this current situation, it is obvious that studying the FAs of wintering ferns is a very urgent task. Therefore, we decided to study changes in the total lipid fatty acid composition of ferns with the season.

## 2. Materials and Methods

### 2.1. Plant Materials

The object of the study was the fronds of *Asplenium scolopendrium* (L.) Newman. This fern is 30–80 cm tall and has a short rhizome covered with scales on the top (Appendix A). The fronds are dark green, leathery, bare, and intact, which is rare among ferns; their length is 10–60 cm, and their width is 3–6 cm. The petioles are shorter than the frond blade [44,45] (Appendix A). On the abaxial surface of the fronds, along the lateral veins, the sori are arranged parallel to each other and covered with two indusia. Their arrangement is reminiscent of Myriapoda legs, from which the species name *scolopendrium* was derived; in Latin, it means “centipede” (Appendix A).

Plant specimens were cultivated in open ground (55°69′85′′ N 38°91′16′′ E) under similar semishaded conditions. In temperate climates, *A. scolopendrium* fronds are evergreen, i.e., they do not die off in winter but remain under snow cover and vegetate the next year. For this study, 3–5 fronds were taken from three different plants three times per year: in spring (April 1–10, after the snow had completely melted and the average daily temperatures were stable above 0 °C), in summer (July 20–30, the period with the longest days) and in autumn (November 1–10, in the absence of permanent snow cover but with temporary snow cover possible; the average daily temperature was stable at below 0 °C). The average monthly temperatures during the experiment (2018–2020) were recorded by using an EClerk-M-T-a automatic thermometer (Relsib, Russia), and the average monthly precipitation was calculated according to http://www.pogodaiklimat.ru (accessed on 6 February 2022) (Figure 1).

### 2.2. Lipid Extraction

Freshly collected fern leaves were weighed and fixed in boiling isopropyl alcohol (CH_3_CH(OH)CH_3)_ for 1 h to preserve the native lipid composition. At a preliminary stage, surface waxes were removed with chloroform (CHCl_3_). The samples were stored at +4 °C until extraction. To extract total lipids, the CH_3_CH(OH)CH_3_ was separated through a Schott glass filter into a 250 mL volumetric flask, and the plant tissue was homogenized in a porcelain mortar and extracted. Lipids were extracted three times sequentially in CHCl_3_/methyl alcohol (CH_3_-OH)/water (H_2_O) (30:20:1.7, by vol.) and CHCl_3_/CH_3_-OH/nitric acid (HNO_3_) (20:10:0.1, by vol.). All the extracts were added to the volumetric flask, and the total volume was adjusted to 250 mL with CH_3_CH(OH)CH_3_ containing 0.001% butylated hydroxytoluene (Sigma-Aldrich, Burlington, MA, USA, 34750) as an antioxidant. The extracts were stored in a refrigerator [46].

### 2.3. Preparation of Fatty Acid Methyl Esters

FA methyl esters (FAMEs) were prepared according to a previously described method [47]. The sample was evaporated to dryness using a rotary evaporator under standard conditions (at not higher 50 °C and 9.5 × 10^−1^ kgf/cm^2^). After saponification was conducted in a boiling solution of 4% sodium hydroxide (NaOH) in CH_3_OH/H_2_O (1:1, by vol.). Then, the sample was evaporated to dryness. Approximately 2 mL of water was added to the dried sample, and unsaponifiable FAs were washed several times with n-hexane (C_6_H_14_) until they became clear. Then, FAs were extracted six times with 2 mL of C_6_H_14_ in acidic conditions (acidifier 20% sulphuric acid (H_2_SO_4_)). The collected C_6_H_14_ was evaporated, 3 mL of CH_3_OH and a 750 μL of acetyl chloride (CH_3_COCl) were added to the sample, and the sample was boiled for 1 h. Then, the sample was again evaporated, 2 mL of H_2_O was added, and FAMEs were extracted six times with C_6_H_14_ under acidic conditions (acidifier 20% H_2_SO_4_). After that, the hexane was evaporated, and 500 μL of benzene (C_6_H_6_) was added [48].

The extract was pipetted onto a chromatographic plate, and a mixture of C_6_H_14_/diethyl ether (C_2_H_5_-O-C_2_H_5_)/glacial acetic acid (CH_3_COOH) (8:2:0.1, by vol.) was used as the mobile phase. When the front reached the top of the plate, the plate was removed and air-dried for 1–2 min. Then, the plate was treated with a 0.001% solution of 2′,7′-dichlorofluorescein in ethanol (C_2_H_5_OH) and air-dried for 5–7 min. The zones with FAMEs were visualized in UV light (λ = 365 nm). Then, the silica gel from the FAME zone of the chromatographic plate was transferred to a Schott glass filter, and the FAMEs were eluted from the sorbent by washing with C_6_H_14_ six times [49].

The FAMEs were analyzed by gas chromatography-mass spectrometry (GC-MS) on an Agilent 7890A GC with a quadrupole mass detector Agilent 5975C fitted with a 60-m capillary column DB-23 (inner diameter 0.25 mm, thickness of stationary phase-(50%-cyanopropyl)-methylpolysyloxane—250 µm). The prepared FAMEs were separated under the following conditions: carrier gas, helium at 1 mL/min; sample volume, 1 µL; split ratio, 4:1 (in numerous analyses, splitless injection was used); evaporator temperature, 260 °C. The oven temperature program was as follows: from 130 to 170 °C at 6.5 °C/min, to 215 °C at 2.75 °C/min (25 min hold at this temperature), to 240 °C at 40 °C/min (30 min hold at 240 °C). The operational temperature of the mass detector was set to 240 °C, and the ionization energy was set to 70 eV. To identify individual FAME species (Appendix A), NIST and Wiley search libraries and ChemStation software were used, and the relative retention time and equal chain length (ECL) value were calculated for each peak [50].

### 2.4. Data and Statistical Analyses

To characterize the unsaturation level of lipid FAs, the unsaturation index (UI) [51] and double-bond index (DBI = (2 × %18:2 + 3 × %18:3)/(%16:0 + %18:0 + %18:1) [52,53] were calculated. The activity efficiencies of the Δ9-, Δ6-, and Δ3-desaturases were calculated as the palmitic desaturation ratio (PDR = %16:1/(%16:0 + %16:1)), palmitoleic desaturation ratio (PlDR = (%16:2 + %16:3)/(%16:1 + %16:2 + %16:3)), palmitolinolenic desaturation ratio (PlnDR = %16:3/(%16:2 + %16:3)), stearic desaturation ratio (SDR), oleic desaturation ratio (ODR), and linoleic desaturation ratio (LDR) [8]. Desaturation and elongation partitioning ratios were calculated based on the ratios of the following FAs: C16 desaturation— %16:1/(%16:0 + %18:0 + %18:1 + %18:2 + %18:3 + %20:0 + %20:1 + %22:0 + %22:1); C18 desaturation—(%18:1 + %18:2 + %18:3 + %20:1 + %22:1)/(%18:0 + %20:0 + %22:0); C16 elongation—(%18:0 + %18:1 + %18:2 + %18:3 + %20:0 + %20:1 + %22:0 + %22:1)/(%16:0 + %16:1) [54].

All experiments were performed in 3–5 replicates with at least three independent executions. The data are presented in tables and a graph as the means ± SEMs. Statistical analysis was performed as described by us previously [8]. Different symbols or letters indicate significantly different values. Mean values were considered significantly different at *p* ≤ 0.05.

## 3. Results and Discussion

Throughout the period of material collection, the major FAs in the total lipids of *A. scolopendrium* were palmitic (16:0), linoleic (9,12-18:2), and α-linolenic (9,12,15-18:3) acids; moreover, the amount of 16:0 decreased significantly during the year, and the amounts of 9,12-18:2 and 9,12,15-18:3 increased in summer and autumn (Table 1). Cold-resistant genotypes of rice are characterized by the accumulation of 9,12-18:2 acids after such exposure when the content of 16:0 decreases and the opposite changes occur in thermophilic genotypes [33]. Thus, the *A. scolopendrium* FA composition changes in autumn (Table 1), as in cold-resistant species. However, a decrease in the level of 9,12,15-18:3 after wintering may indicate dehydration of the plant, since for species that are less resistant to drought, a decrease in the level of this acid during tissue dehydration has been shown [55,56]. In addition, the decrease in the amount of 9,12,15-18:3 may be because this FA, released from membrane lipids by means of regulated lipase activity, is the precursor molecule for phyto-oxylipin biosynthesis [57]. Oxylipins can play an important role in the resistance of plants to pathogens [58,59], which is especially important when plants are emerging from a wintering state.

In spring, oleic acid (9-18:1) was also a major FA (Table 1). It is known that 9-18:1 can stimulate plasma membrane phospholipase D, which generates phosphatidic acid to attenuate H_2_O_2_-induced cell death [60,61]. At the same time, peroxide is one of the main reactive oxygen species (ROS) that plants encounter when exposed to lows temperatures [29,62], and larger amounts of 9-18:1 in *A. scolopendrium* in the spring may indicate its role in the antioxidant defense of the fern. In autumn, the major FAs in *A. scolopendrium* include arachidonic acid (5,8,11,14-20:4), which is not found in the tissues of flowering plants [63,64]. However, this FA is characteristic of lipids in algae [65], mosses [66,67], and other ferns [8,68,69].

Thus, among the major FAs, there was a tendency for the accumulation of unsaturated FAs, especially polyunsaturated FAs, during the growing season, with a decrease in the level of saturated FAs. A similar trend was also observed for minor FAs: the levels of myristic (14:0), pentadecanoic (15:0), and stearic (18:0) acids decreased significantly, while the levels of 7,10-hexadecadienoic (7,10-16:2) and 7,10,13-hexadecatrienoic (7,10,13-16:3) acids increased in a stepwise manner (Table 1). Such tendencies in the *A. scolopendrium* FA composition could affect the level of total lipid saturation. In fact, we observed that two integral indicators of the level of lipid unsaturation and the number of double bonds (the UI and DBI, respectively) increased significantly during the growing season (Figure 2). The ratio of the levels of saturated and unsaturated fatty acids underlies the regulation of the phase transition point of biological membranes [57]. The presence of a large number of unsaturated FAs in the membrane, especially of polyunsaturated FAs, the proportion of which in *A. scolopendrium* increased as the cold season approached (Figure 3), leads to increased membrane fluidity. Therefore, the higher the UI and DBI values are, the more resistant the cell is to low temperatures [70,71]. Thus, changes in the levels of FAs systematically prepare *A. scolopendrium* for successful wintering.

It is known that unsaturated FAs are formed from saturated FAs with the help of enzymes desaturases [38]. The activity of all enzymes, as protein molecules, depends on temperature [72,73]. Our research showed that the activities of only Δ6-desaturases (PlDR, ODR) and Δ3-desaturases (PlnDR, LDR) were significantly higher in summer and autumn (Figure 4A). At the same time, the desaturation process was significantly intensified only for 18-FAs, against the background of a monotonous increase in the level of 16-FA elongation (Figure 4B). Thus, in summer and spring, active synthesis of 9,12-18:2 and 6,9,12-18:3, which are precursors of 5,8,11,14-20:4 and eicosapentaenoic acid (5,8,11,14,17-20:5), respectively, occurs in the tissues of *A. scolopendrium* fronds [74,75]. Therefore, the presence of 5,8,11,14,17-20:5 in the tissues of *A. scolopendrium* was notable, the absence of which we previously noted in some thermophilic ferns [8]. The abundance of this FA was not very high, but it increased significantly in autumn (Table 1). Thus, significant contributions to the increase in UI and DBI values in autumn (Figure 2) were observed for tetra- and pentaenic very-long-chain FAs (VLCFAs), particularly 5,8,11,14-20:4 and 5,8,11,14,17-20:5 (Figure 3). It is obvious that preparation for the winter period via the accumulation of unsaturated FAs begins in the summer, and it is believed that the accumulation of polyunsaturated VLCFAs can prevent membranes from undergoing overfluidization in the warm period [76].

The presence of capric (10:0), lauric (12:0), and tridecanoic (13:0) acids, also referred to as medium-chain FAs (MCFAs) [77,78,79], can be considered a distinctive feature of the FA composition of lipids in *A. scolopendrium* fronds after wintering (Table 1). MCFAs are formed de novo during synthesis via the participation of two enzyme systems, acetyl-CoA carboxylase and fatty acid synthase, the end products of which are 16:0 and 18:0 [80,81]. Thus, the appearance of MCFAs among *A. scolopendrium* lipids indicates the existence of active FA synthesis processes in spring. This is likely due to the restoration of the photosynthetic apparatus of the thylakoids after wintering, as it is known that de novo biosynthesis of FAs plays a critical role in the response of the photosynthetic apparatus to low temperature [82].

In addition, MCFAs are agonists of peroxisome proliferator-activated receptors [83]. Unfortunately, there is no such information available for plants. However, it is known that wintering can lead to damage to lipids [84,85] during oxidative stress [86,87,88], with ROS representing one type of factor involved in this process [89,90]. Based on these data, it can be generally stated that *A. scolopendrium* MCFAs can participate in the modulation of peroxisome activity, which is an integral part of cellular protection against oxidative damage caused by ROS [91,92].

In contrast, in the summer and autumn, MCFAs were absent from the *A. scolopendrium* lipids. However, new species of VLCFAs, namely, pentacosylic (25:0), cerotic (26:0), and montanic (28:0) acids, which were not observed in spring, appeared. Their presence did not lead to any notable changes in the total number of VLCFAs (Figure 2). This likely supports the hypothesis that the ΣVLCFA levels remain constant even as new species of FAs emerge, which may be because an increase in their level would lead to the suppression of cell proliferation [93] and accordingly could slow the growth of *A. scolopendrium*. It can be assumed that new types of VLCFAs appear as a consequence of activation of the synthesis of epicuticular waxes, which are the substrate for the biosynthesis of such FAs [94,95]. It is logical to assume that the impregnation of cuticle waxes can protect fern fronds from critical water loss in the winter.

## 4. Conclusions

In summary, the results suggest that in addition to other factors [26] associated with stability mechanisms, FAs play a very important role in the resistance of *A. scolopendrium* to low and subzero temperatures. Because of the presence of polyunsaturated VLCFAs, which are not characteristic of flowering plants (5,8,11,14-20:4 and 5,8,11,14,17-20:5 [96]), ferns adapt their membrane composition to frost, and as a result, they are likely able to grow at moderate latitudes and even closer to the poles, although they initially inhabited only places with a warm climate. At the same time, the FAs probably not only prepare *A. scolopendrium* for winter but also aid in the emergence of the fronds from snow. Since MCFAs were detected at the beginning of the spring, they are likely actively involved in the antioxidant protection of photosystems (along with 9-18:1) and in the repair of photosystems after exposure to cold stress.

All of the above findings suggest that the FA composition in ferns, which previously could be considered archaic in evolutionary terms, provides many hidden opportunities for the adaptation of *Polypodiophyta* and is a fundamental component of the ecological plasticity of these plants.

## Figures and Tables

**Figure 1 biology-11-00507-f001:**
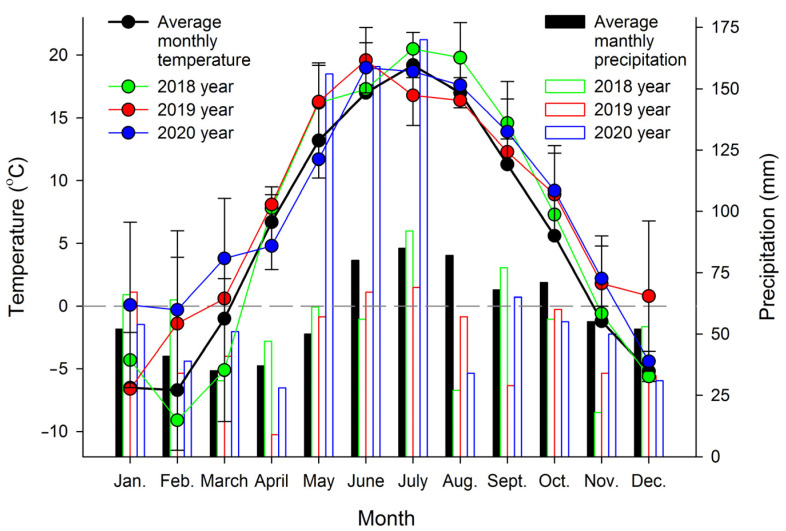
Monthly averages of temperature and precipitation at the *Asplenium scolopendrium* collection site during 2018–2020. Values are presented as the means ± SEMs.

**Figure 2 biology-11-00507-f002:**
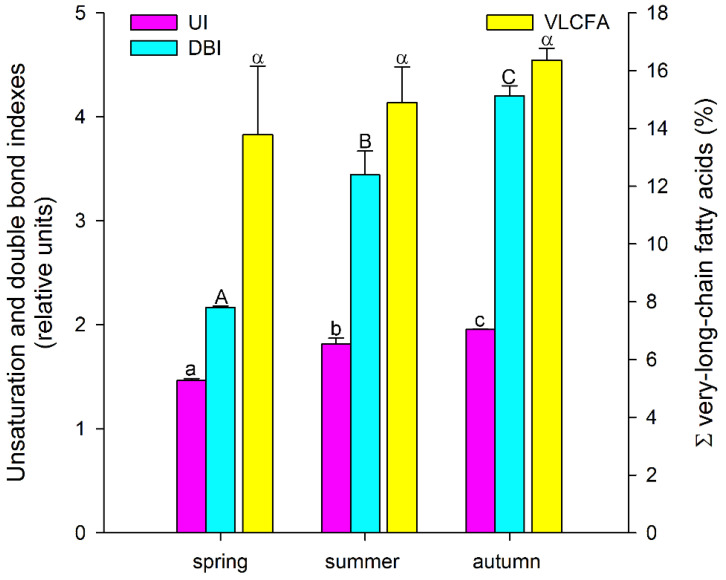
Unsaturation index (UI) and double-bond index (DBI), as well as the total amount of very-long-chain fatty acids (ΣVLCFAs) in *Asplenium scolopendrium* fronds in spring, summer, and autumn. Values are presented as the means ± SEMs. Different letters indicate a significant difference between the means (*p* ≤ 0.05). One-way ANOVA, followed by Tukey’s HSD test, was performed separately for each parameter.

**Figure 3 biology-11-00507-f003:**
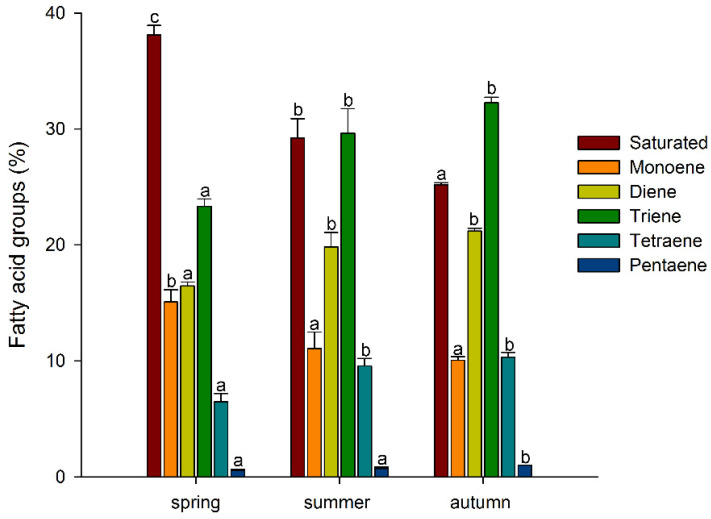
Percentage of saturated and unsaturated (monoene, diene, triene, tetraene, and pentaene) fatty acid components in *Asplenium scolopendrium* fronds in spring, summer, and autumn. Values are presented as the means ± SEMs. Different letters indicate a significant difference between the means (*p* ≤ 0.05). One-way ANOVA, followed by Tukey’s HSD test, was performed separately for FA groups.

**Figure 4 biology-11-00507-f004:**
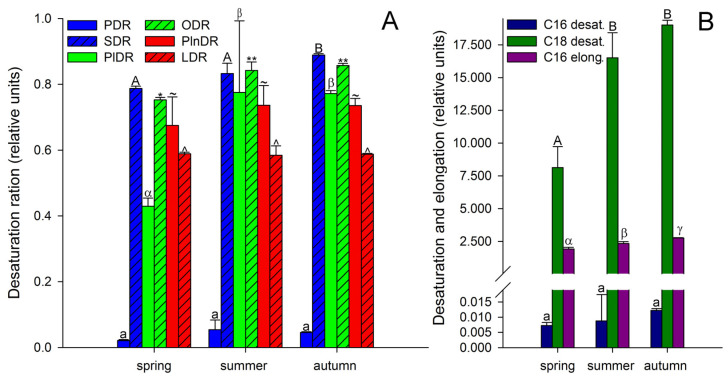
Desaturation ratios (**A**) and desaturation and elongation (**B**) of *Asplenium scolopendrium* fronds in spring, summer, and autumn. Abbreviations: *PDR*—palmitic desaturation ratio, and *SDR*—stearic desaturation ratio (the activity efficiencies of the Δ9 desaturases); *PlDR*—palmitoleic desaturation ratio, and *ODR*—oleic desaturation ratio (the activity efficiencies of the Δ6 desaturases); *PlnDR*—palmitolinolenic desaturation ratio, and *LDR*—linoleic desaturation ratio (the activity efficiencies of the Δ3 desaturases); *C16 desat.*—C16 desaturation; *C18 desat.*—C18 desaturation; *C16 elong.*—C16 elongation ratio. Values are presented as the means ± SEMs. Different letters or symbols indicate a significant difference between the means (*p* ≤ 0.05). One-way ANOVA, followed by Tukey’s HSD test, was performed separately for each parameter.

**Table 1 biology-11-00507-t001:** Lipid fatty acid composition of *Asplenium scolopendrium* fronds in spring, summer, and autumn (mass % of the amount of FAMEs).

Fatty Acid	Spring	Summer	Autumn
10:0	0.1 ± 0.1 b	* a	* a
12:0	0.3 ± 0.0 b	* a	* a
13:0	0.1 ± 0.0 b	* a	* a
14:0	1.4 ± 0.1 c	0.6 ± 0.2 b	0.2 ± 0.0 a
15:0	0.6 ± 0.1 c	0.4 ± 0.1 b	0.2 ± 0.0 a
16:0	27.5 ± 1.6 c	23.4 ± 1.7 b	20.0 ± 0.3 a
4-16:1	0.6 ± 0.3 b	* a	* a
7-16:1	0.6 ± 0.1 a	0.7 ± 0.7 a	1.0 ± 0.1 a
9-16:1	1.4 ± 0.8 a	1.6 ± 1.0 a	0.3 ± 0.0 a
14-16:1	0.1 ± 0.1 b	* a	* a
7,10-16:2	0.2 ± 0.1 a	0.5 ± 0.1 b	0.9 ± 0.1 c
7,10,13-16:3	0.3 ± 0.0 a	1.6 ± 0.4 b	2.4 ± 0.1 c
18:0	3.2 ± 0.2 a	1.7 ± 0.2 c	1.0 ± 0.0 b
9-18:1	11.9 ± 0.2 b	8.4 ± 1.5 a	8.0 ± 0.3 a
11-18:1	* a	0.4 ± 0.4 ab	0.6 ± 0.0 b
9,12-18:2	14.9 ± 0.3 a	18.5 ± 1.1 b	19.8 ± 0.2 b
10,13-18:2	0.8 ± 0.1 b	0.5 ± 0.4 ab	0.1 ± 0.1 a
6,9,12-18:3	0.3 ± 0.1 a	0.7 ± 0.1 c	0.5 ± 0.1 b
9,12,15-18:3	21.4 ± 0.6 a	26.1 ± 1.6 b	28.2 ± 0.4 b
19:0	0.2 ± 0.2 a	0.1 ± 0.1 a	0.1 ± 0.0 a
20:0	0.8 ± 1.0 a	0.4 ± 0.4 a	0.7 ± 0.0 a
11-20:1	0.1 ± 0.2 a	0.1 ± 0.1 a	0.2 ± 0.1 a
11,14-20:2	0.6 ± 0.1 b	0.3 ± 0.1 a	0.5 ± 0.0 b
8,11,14-20:3	1.0 ± 0.3 a	1.2 ± 0.6 a	1.1 ± 0.1 a
11,14, 17-20:3	0.4 ± 0.1 b	* a	* a
5,8,11,14-20:4	6.3 ± 0.5 a	9.4 ± 0.6 b	10.3 ± 0.3 b
8,11,14, 17-20:4	0.2 ± 0.2 a	0.1 ± 0.2 a	0.1 ± 0.1 a
5,8,11,14, 17-20:5	0.6 ± 0.1 a	0.7 ± 0.1 a	1.0 ± 0.0 b
21:0	0.1 ± 0.1 a	* a	0.1 ± 0.1 a
22:0	2.2 ± 0.3 b	1.1 ± 0.3 a	1.3 ± 0.1 a
13-22:1	0.5 ± 0.1 b	* a	* a
23:0	0.2 ± 0.1 a	0.2 ± 0.1 a	0.2 ± 0.1 a
24:0	1.1 ± 0.3 a	0.9 ± 0.2 a	0.8 ± 0.1 a
25:0	* a	0.1 ± 0.1 b	0.1 ± 0.0 b
26:0	* a	0.2 ± 0.2 b	0.2 ± 0.1 ab
28:0	* a	0.1 ± 0.1 a	0.1 ± 0.1 a

Values are presented as the means ± SEMs. Different letters indicate a significant difference between the means (*p* ≤ 0.05). One-way ANOVA, followed by Tukey’s HSD test, was performed separately for each FA. * Trace (≤0.1%).

## Data Availability

Not applicable.

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
