# Peer review of "Significance of Lipid Fatty Acid Composition for Resistance to Winter Conditions in Asplenium scolopendrium"

_biology, 2022, doi:10.3390/biology11040507_

Round 1

Reviewer 1 Report

Referee 1:

Comments to Author

This manuscript needs extensive revisions. The author need to read more relevant literature and revised more deeply in the introduction and discussion part. The materials and methods is also unclear. I suggested that following points should be addressed/clarified before the acceptance of the manuscript.

General comment: the English grammar needs correction throughout the manuscript.

  1. The introduction, I think this paragraph can be rewritten deeply and specifically. What are physiological functions of fatty acids in plants? How they function in stress resistance? And it may cause what gene expression changes in plant? They all should be supplemented in this part.
  2. Line 65, “some parameters”----Please describe detailedly.
  3. Line 68, use “and previous studies reported that”to replace “There is no doubt that”.
  4. Line 70, use “Unsaturated FAsaccumulation in cell membranes” to replace “The accumulation of unsaturated FAs in cell membranes” 
  5. Line 72, add “but”before“This aspect...”
  6. Line 73-75, the sentences “It is also known that ...dehydration”should be revised into “Dehydration leads to damage to cell membranes, and FA contributes to better adaptation of the cell to dehydration due to its composition of membrane lipids.”
  7. Line 76, use “current situation”to replace “information” 
  8. Line 79, use “Plant materials”to replace “Research objects” 
  9. Figure S1, why is there no legend on the picture?
  10. Line 80-87, this paragraph is not closely connected to the key point of the manuscript, and I think it can be cut out appropriately.
  11. Line 91, simply collecting 3 plants within a single treatment makes them psuedoreplicates not experimental replicates, and data under this method were unreliable and incorrect.
  12. Line 114-Line 150, in this section, there are too much “Then”. Please consider the fluency and coherence of the sentences in this manuscript to make it easier to read.
  13. Line 121, Line 122, Line 126, Line 127, Line 164, “1-2 ml”, “a few drops of”, and “3-5 replicates”are not accurate .The experimental design and some of the methods are unclear. It is unclear from the manuscript whether the treatments were repeated.
  14. Line 173-174, use specific name of FA to replace “the first FA”, “the second”, and “the third”.
  15. Line 175-177, this sentence should be rewritten.
  16. Line 177-178, only basing on the changes in FA composition during the year to judge whether the plant is a cold-resistant species, which I don’t agree with.
  17. Figure 2, a) does UI have significant differences between in summer and in autumn? Please check it. b) Place the different letters above the error range.
  18. Line 233, Line 235, “Δ6- and Δ3-desaturases”and “18-FAs” were only mentioned in “Materials and method”. In Figure 4, they should be linked to the abbreviations like PDR, ODR, SDR in the figure legend.

Author Response

Dear reviewer, we are grateful for your work with our manuscript. Detailed answers to each question and comment please see the attachment.

Best regards,

Alexander Voronkov and Tatiana Ivanova

Reviewer 2 Report

General

This manuscript measures the composition of lipid fatty acids in a fern subject to winter freezing and shows clear differences in these constituents with season. A key concern for this paper is that the authors make the assumption (possibly correctly) that the fatty acids provide a mechanism of resistance to cold. However, it is still an assumption, and needs to be treated as such in the text. The paper shows a correlation between these two elements but not causation. The link between fatty acid composition and concentration and cold resistance needs further direct proof. 

In the methods, there is no description of the habitat that the sampled plants were growing in. A more detailed description of this habitat is desirable.  Were all specimens sampled growing in homogeneous conditions? Were these plants wild or cultivated? Were they growing in shade or in the open?

Although samples were collected over three years, the factor ‘year’ was not included in the statistical analysis. Were there any differences among years? As well, the statistics include a large number of separate post-hoc tests. This suggests that some Bonferroni type corrections would be applicable to reduce the chance of Type 1 errors.

The form of Figure 3 is difficult to follow. Please consider another type of graph for these data.

Also, note that whether temperatures rise of fall as one moves north or south depends on which hemisphere you live in. Please adjust the language so it makes sense to readers from either hemisphere.

Specific:

Title: the title doesn't accurately represent the scope of the paper. The paper shows clear changes in lipid fatty acid composition with season but does not test whether these constituents have a role in freezing resistance. This is an assumption based on the perceived roles of these compounds in other plants. The current title overstates what can be logically concluded from the studies undertaken.

Line 8: Change ‘northern’ to ‘polar’. Northern latitudes could include anything north of the Equator and high northern latitudes are not cold in the southern hemisphere.

Line 9: ‘more or fewer’ – these contradict each other. Do you mean ‘fewer’.

Line 10: As with line 8, using the term ‘southern’ depends on your location. For example, ‘more southern’ areas in the southern hemisphere are colder. Change ‘southern’ to ‘lower latitude’.

Line 22: Note that ferns are the oldest vascular land plants. Bryophytes arose prior to ferns.

Line 28: How would determining changes in fatty acid concentration infer a role in cold resistance? There is a logical issue here.

Line 32: Showing a change in fatty acid composition in fronds doesn’t necessarily imply adaptation to growing conditions and preparing to overwinter.

Line 38: Are you including lycophytes in ferns here? Usually, true ferns are just the monilophytes with the lycophytes separate.

Line 41: What about the emergence of gymnosperms?

Line 47: What about ferns at high latitudes in the southern hemisphere, e.g., Polystichum vestitum in the subantarctic?

Line 72: State your implied assumption here that the accumulation of unsaturated fatty acids in cell membranes of ferns promotes tolerance to low temperatures.

Line 77: I think ‘extremely urgent’ overstates the importance of this study. It is certainly a desirable line of enquiry but the case for urgency has not been made.

Line 77: Provide an objective statement for this study at the end of the Introduction? What specifically did you hope to achieve and what approach was taken?

Line 82: Suggest change ‘unique’ to ‘rare’. Entire leaves such as these are found in some other species of fern, e.g., Asplenium nidus, so can’t be unique.

Line 88: What does ‘open ground’ mean? Describe the situation that the sampled plants were grown in. Were these plants growing in the wild? If so, what habitat/ecosystem were they growing in? If growing in cultivation, also what habitat were they growing in?

Line 91: Confirm how many times an individual plant was sampled? I assume that previously unsampled plants were sampled at each of the nine sampling times. Is this correct?

Figure 1: This figure presents data collected during your study so should be in the Results section, not in the methods. You might also consider putting this in the supplementary information section if possible.

Line 103: How long were these fronds kept between collection and weighing?

Line 108: What were the ‘slight modifications’ to the methods indicated?

Line 116: Again, what were the ‘slight modifications’ referred to here?

Line 167: One-way ANOVA was used for the analyses but there were two possible factors: season and year. Season appears to be the factor used for the analyses, but how was year accounted for in your statistics? Was there any change in fatty acids per year? Was any year notably colder or warmer than the others, and did fatty acid composition change accordingly?

Line 175: Change ‘plant species’ to ‘rice’ as this comment is based on reference 27.

Line 177: This assumes that A. scolopendrium would behave the same as rice. Note this assumption.

Table 1: Locate at the end of a paragraph. This table could also be considered for the supplementary information section. With so many individual post-hoc tests in this table, a Bonferonni correction (or a Holm-Bonferonni correction or similar) is also appropriate.

Figure 3: I suggest you present this figure as a 100% stacked column chart, or a vertical bar chart. I find the circular format of this graph difficult to follow and particularly to compare changes within categories. As well, your caption needs to indicate that these are percentages not absolute values.

Line 234: Indicate which bars you refer to in Fig 4A in this sentence.\

Fig. 4: In the caption, indicate what the acronyms in the legend refer to, e.g., PDR, SDR.

Line 259: A. scolopendrium misspelt (change capital S to lower case).

Line 274: Presumably the ‘new species’ referred to here are different types of lipid fatty acids. I suggest you use another term to describe different types of chemicals as ‘species’ could be misconstrued to mean biological species.

Line 280-282: Delete sentences that seem to come from author guidelines.

Line 284-286: This is overstated. Note that your study has only measured the composition and abundance of FAs in fronds of this species but have not tested their role in cold tolerance (even though it is likely).

Line 289: Replace ‘farther north’ with ‘further towards the poles’. In the southern hemisphere, farther north is towards warmer climates.

Lines 290-294: Again, you have identified the existence of these compounds but not their roles. This section is based on assumptions of the roles of these compounds and should be written in this context.

References: several species names in the references need to be in italics.

Author Response

(The authors gave the same response as above.)

Round 2

Reviewer 1 Report

Comments:

    1. Table 1: It is recommended that the label shape of the line chart in Figure 1 be marked differently. The graphic display is not very clear, it is suggested to use solid columns to highlight the average value.
    2. Please check if there is any problem with the calculation of the average in Figure 1, it should show in the legend that the average is the average of several days.
    3. The chemical formula in the text should be written with its name, such as: isopropanol (СН3СH(OH)CН3). Please check the full text for similar situations.
    4. Line 129: Does the saponification reaction take place in a rotary evaporator? This should be clearly stated. Is the 4% NaOH solution boiling or cooling?
    5. Line 130: The temperature and mode of evaporation should be specified.
    6. Line 132: The NaOH solution was added in the previous step, and then it was extracted with n-hexane in an acidic environment. How did the acidic environment come about? The extraction method and volume of the 6 extractions were not clearly explained.
    7. In 2.3 Preparation of fatty acid methyl esters: The description of the FA methylation method is not detailed enough, please rewrite this part. Volume is a strict dosage, do not use inaccurate terms like About and 1-2 ml.
    8. Line 171: What does 3-5 replicates mean? Experiment replication should be clear.
    9. Table 1: The notes should be placed below the table.
    10. Why did not collect the experimental materials in winter? Your topic is resistance to winter conditions, then winter fatty acid changes should be the focus, which is missing in the article.
    11. Line138-145: The experimental results in this part are suggested to be added to the article.
    12. The GC-MS chromatogram is suggested to be added to the supplementary file.。

Author Response

Dear reviewer, we have corrected the manuscript, the answers to your questions Please see the attachment.

Round 3

Reviewer 1 Report

The manuscript has been revised in accordance with the revision requirements and is being considered for publication.